# Pharmacist-Managed Therapeutic Drug Monitoring Programs within Australian Hospital and Health Services—A National Survey of Current Practice

**DOI:** 10.3390/pharmacy10050135

**Published:** 2022-10-18

**Authors:** Paul Firman, Ken-Soon Tan, Alexandra Clavarino, Meng-Wong Taing, Karen Whitfield

**Affiliations:** 1School of Pharmacy, The University of Queensland, Woolloongabba, QLD 4102, Australia; 2School of Medicine, Griffith University, Gold Coast, QLD 4222, Australia; 3Faculty of Medicine, The University of Queensland, Herston, QLD 4006, Australia; 4School of Public Health, The University of Queensland, Saint Lucia, QLD 4072, Australia

**Keywords:** clinical pharmacy, hospital pharmacy, therapeutic drug monitoring, pharmacy expanded scope

## Abstract

Pharmacist-managed therapeutic drug monitoring (TDM) services have demonstrated positive outcomes in the literature, including reduced duration of therapy and decreased incidence of the adverse effects of drug therapy. Although the evidence has demonstrated the benefits of these TDM services, this has predominately been within international healthcare systems. The extent to which pharmacist-managed TDM services exist within Australia, and the roles and responsibilities of the pharmacists involved compared to their counterparts in other countries, remains largely unknown. A cross-sectional online survey was conducted evaluating pharmacist-managed TDM programs within Australian hospital and healthcare settings. Pharmacist perceptions were also explored about the strengths, weaknesses, opportunities, and barriers associated with implementing a pharmacist-managed TDM service. A total of 92 surveys were returned, which represents a response rate of 38%. Pharmacist-managed TDM programs were present in 15% of respondents. It is only in the minority of hospitals where there is a pharmacist-managed service, with pharmacists involved in recommending pathology and medication doses. The programs highlighted improved patient outcomes but had difficulty maintaining the educational packages and training. For hospitals without a service, a lack of funding and time were highlighted as barriers. Based on the findings of this survey, there is minimal evidence of pharmacist-managed TDM models within Australian hospital and health services. A standardized national approach to pharmacist-managed TDM services and recognition of this specialist area for pharmacists could be a potential solution to this.

## 1. Introduction

There have been recent discussions in Australian hospitals about the opportunity for the role of pharmacists to be expanded in therapeutic drug monitoring (TDM) [1]. Compared to the United States, New Zealand and the United Kingdom, the pharmacist’s role in TDM in Australia has largely been restricted to providing advice about the most appropriate time to perform TDM and the interpretation of the results. Greater involvement and expansion of the role to include ordering medication-related pathology (i.e., drug levels), for example, has been largely unexplored, and is in part due to current regulations within Australia [2,3]. 

In other jurisdictions, such as New Zealand, the prescribing pharmacist’s framework indicates they have the authority to order and interpret laboratory tests within a collaborative health team environment, although there is a lack of research describing the extent of this practice [4]. In the United Kingdom, pharmacists can order laboratory tests if they are either an independent or supplementary prescribing pharmacist. A recent survey of pharmacists in the United Kingdom reported 50% of respondents routinely ordered pathology for TDM [5]. In the United States (US), the majority of primary care pharmacists have the authority to order laboratory tests within a collaborative practice agreement with a medical officer or provide point-of-care testing and pharmacy education within a clinic or pharmacy setting [6,7]. Within the US hospital system, a survey by The American Society of Health-System Pharmacists (ASHP) reported that 97.3% of hospitals had pharmacists routinely monitor serum medication concentrations or their surrogate markers, 85.5% authorized pharmacists to order an initial serum medication concentration, and 84.5% allowed dosage adjustments of medications being monitored by pharmacists [8]. 

Medication management is becoming more complex and, with greater emphasis on individualized care, there is scope for greater pharmacist involvement in TDM to include ordering and interpreting laboratory tests and pharmacist prescribing [9,10,11]. Evidence also suggests the integration of pharmacy services in TDM can impact positively on patient care, with improved sampling time and clinical interpretation of drug levels leading to more efficient and cost-effective delivery of care [12,13]. There is evidence to suggest that, when pharmacists are directly involved in ordering pathology and dose adjustments, substantial cost savings can be demonstrated by reducing inappropriate requests and increasing the optimization of initial dosing for medications such as antibiotics [12,13,14,15]. Although evidence has demonstrated the benefits of these TDM services, the extent to which healthcare settings in Australia have made use of the knowledge and skills of pharmacists to provide high-level, patient-focused services within this area is not known [16,17].

The aim of this study was to identify the extent to which pharmacist-managed TDM services exist within Australia and the roles and responsibilities of the pharmacists involved. The study also explored the perceptions of hospital and health service pharmacists about any strengths, weaknesses, opportunities, and barriers associated with implementing a pharmacist-managed TDM service.

## 2. Materials and Methods

### 2.1. Study Design

A cross-sectional online survey was conducted, which aimed to determine the presence or otherwise of pharmacist-managed or pharmacist-led TDM services within hospital and health services in Australia. The survey explored the presence of such a program, the scope of practice, medications included in the program, operational requirements, attitudes, and practices of pharmacists towards the service, as well as any perceived barriers and enablers associated with the establishment and delivery of these services. The instrument was developed from the literature [18,19,20]. This included literature on surveys conducted within the area of TDM from an international perspective and surveys on pharmacist confidence when working in an extended scope position [21].

The survey was piloted by 10 independent health professionals (pharmacists, nurses, and medical officers) across hospitals and educational institutions (universities) for content validity prior to its distribution. The main feedback related to demographic data and the descriptors for the digital capabilities of the hospital or healthcare facility, and some wording and definitions around specific terminology, including “pharmacist-managed TDM”. There was also feedback provided on whether participants thought this should be a role of the pharmacist, or whether it was beyond their scope. Feedback also suggested more opportunity to provide free-text comments in some specific questions (including the specific roles of pharmacists within TDM, and enablers and barriers to the programs within facilities) and a general comments section at the conclusion of the survey. 

### 2.2. Measures

The survey (Appendix A) was divided into three sections. Section one focused on the demographic details of the participating hospitals; section two focused on current issues around TDM and roles and responsibilities; and section three focused on current practices around TDM within the hospital, whether a pharmacist-managed TDM program existed, and whether there were any enablers or barriers. The survey consisted of a combination of direct answer questions and questions using Likert scales with five response categories (1 = never, 2 = rarely, 3 = sometimes, 4 = often, and 5 = always). Provision was also made within the survey for additional responses and commentary from participants to allow any additional information to be gathered.

#### 2.2.1. Demographic Data

The demographic section collected information on location (including state or territory and metropolitan or rural), size of the hospital based on bed numbers, and whether the hospital utilized an integrated electronic medical record or a paper-based system. 

#### 2.2.2. Current Roles and Issues within TDM

This section comprised questions relating to which specific health professionals within the hospital and health service had a role in the current TDM process, and any issues experienced within the TDM process. 

#### 2.2.3. Current Practice around TDM and the Presence of a Pharmacist-Managed TDM Service

This section focused on the presence, or not, of a pharmacist-managed TDM program. If the hospital operated this service, it included questions on what was included within the service, such as the role of the pharmacist, medications, education and training requirements, and the benefits or risks of such a service. If the hospital did not offer the service, it explored the reasons behind this and any barriers. 

This study received ethical approval from the institution’s ethics committee as per the National Health and Medical Research guidelines [22].

### 2.3. Participants

The survey was distributed to Directors of Pharmacy in both public and private hospitals across all Australian states and territories via email in June 2020 via a web-based platform. The Directors of Pharmacy were identified from a list compiled by the Society of Hospital Pharmacists Australia (SHPA). This list comprised Directors of Pharmacy from both the public and private hospital sectors and from metropolitan, regional, and rural sites across Australia. It included more than 90% of all hospital pharmacy departments across Australia. Respondents were invited to complete the survey on behalf of their department; alternatively, they could invite a delegate from within the pharmacy department to complete it on the department’s behalf or decline the invitation. 

### 2.4. Data Collection

The survey used the electronic internet platform Checkbox to collect all data. The survey was available for four weeks with an initial invitation sent at the start of the survey and one reminder email sent two weeks after the initial distribution. After four weeks, the survey was closed. All invitees were sent a unique link to the survey to ensure multiple responses from a single site were not received. Participation was voluntary and no compensation was offered for completion of the survey.

Two researchers reviewed the free-text comments received in the survey and grouped these into themes. These themes were discussed with the wider research team and adjusted based on consensus. 

### 2.5. Data Analysis

Analysis was performed using the Stata statistical software package (Version 15, StataCorp LLC, College Station, TX, USA). Descriptive statistics with frequencies and mean ± standard deviation were used where appropriate. Percentages were calculated based on the number of respondents who answered each question.

The survey took approximately 15–20 min to complete. 

## 3. Results

The survey was distributed to 242 Directors of Pharmacy across Australia from the 15 June to the 12 July 2020. A total of 92 surveys were returned, which represents a response rate of 38%. 

### 3.1. Demographic Characteristics

The respondents were from every state and territory within Australia (Table 1). There was representation from both metropolitan, regional, rural, and remote public hospitals (42%) and the remainder were from private metropolitan hospitals (16%), all with varying numbers of acute inpatient beds (Table 1). There was an even distribution of hospitals which utilized an integrated electronic medical record system, those that utilized a completely paper-based system, and those that used a combination of both (e.g., electronic medication prescribing and a paper-based notes system, or electronic systems in specific areas such as the intensive care unit) (Table 1). From 50 participant responses, 86% did not have access to a clinical pharmacology service, while the remainder had a service either on or off site. There was no correlation between the type or digital infrastructure of the hospital. There was a strong correlation between TDM services offered and the size of the hospital, with larger hospitals (501–1000) more likely to offer these pharmacist-managed TDM services. 

### 3.2. Therapeutic Drug Monitoring

Australian pharmacists rarely ordered pathology for TDM purposes (13% of respondents). Medical officers most frequently ordered pathology (80% of respondents) and nurses least often (4% of respondents). Medical officers were also most likely to take responsibility for the review and interpretation of drug levels (50%), followed by pharmacists (31%) and nurses (12%) (Table 2). Table 3 provides information about the main issues associated with effective TDM as perceived by pharmacists. There was an even distribution of issues associated with the incorrect ordering of pathology, sample collection and inappropriate interpretation of the drug level. 

### 3.3. Pharmacist-Managed Therapeutic Drug Monitoring Programs

In total, fifteen percent of respondents reported having a pharmacist-managed therapeutic drug monitoring program within their facility. None of the pharmacist-managed TDM programs provided a comprehensive service covering all medications. The most common groups of medications managed by the pharmacist service were glycopeptide antibiotics and aminoglycoside antimicrobials (*n* = 10 or 76% of all programs), followed by digoxin (*n* = 3 or 23% of all programs) and anti-epileptics (*n* = 2 or 15% of all programs) (Table 4). 

Table 5 describes the specific roles performed by pharmacists in the pharmacist-managed TDM program. A range of activities were undertaken by the pharmacist, including ordering pathology; however, prescribing subsequent doses of medication rarely occurred (average Likert score of 1.9) (Table 5). Approximately 70% (10/13) of the pharmacist-managed TDM programs required staff to complete a credentialling or education package in order to participate. Table 6 describes the implementation of an education or credentialling package within the department and associated stakeholders. 

### 3.4. Themes

A number of overarching themes were identified from the free-text comments, which included medication safety, resourcing and funding, and governance/socio-political matters. Medication safety comments included more appropriate utilization of pharmacist skills, taking ownership of medication safety for high-risk medications, and the opportunity for better patient outcomes. Comments by respondents included:


*“It is an important area that many doctors and nurses are not confident in so there are potential benefits in terms of patient outcomes”*



*“I think a position like this is important—we are the medication experts/specialists and having pharmacists’ roles like this should help take ownership of medication safety”.*


In the governance and socio-political theme, issues were raised around the importance of TDM and in prescribing and administering medications. Comments referred to the credibility of structured roles, greater engagement with the multidisciplinary team (MDT), better use of hospital resources, deskilling of other staff (i.e., medical officers), and the lack of a policy for such a program. One respondent commented: 


*“Pharmacist led TDM requires clear hospital policy so that doses may be adjusted with confidence by pharmacists with most sites lacking policy and therefore assertive pharmacists to lead a TDM program”*


Finally, resourcing and funding issues included added workload for pharmacists, time to deliver the service, lack of funding, department resources, e.g., regional/rural and remote services vs. metropolitan services, and difficulty maintaining credentialling and education packages. One respondent commented: 


*“I would be happy to have a pharmacist-led TDM service but at a regional/rural hospital, there isn’t often the volume of patients requiring TDM or the level of acutely unwell that some for the larger hospitals would have. Having said that, my view is that many doctors place a low priority on TDM, and it isn’t often used when clinically indicated.”*


## 4. Discussion

This study details the current extent of pharmacist-managed TDM services in the Australian hospital and healthcare system. A total of 15% of the surveyed sites reported having a pharmacist-managed TDM program within their facility. The role of the pharmacist within the program consisted of alerting the medical officers when TDM was to occur and calculating subsequent doses for medications. In only a small number of facilities did pharmacists request relevant pathology or actively prescribe subsequent doses. 

Within Australian hospitals currently, medical staff and pharmacists will routinely refer to approved nomograms and protocols when they evaluate drug concentrations for TDM. For more complex cases, they may refer to experts, including clinical pharmacologists, an infectious disease physician or an antimicrobial stewardship pharmacist for anti-infective medications or medical staff within the laboratory where the assay was performed. Unfortunately, in a recent Australian survey, it was reported that one-quarter of Australian hospitals lack endorsed guidelines for anti-infective TDM and that dose prediction software was only accessible in 51% of Australian hospitals [23].

An effective and successful TDM process requires a multidisciplinary approach to provide an efficient service and ensure clinically meaningful drug concentrations are achieved [24,25,26,27]. This survey found medical officers were the most common member of the team involved in TDM within Australian hospitals, and were responsible for ordering pathology, as well as reviewing and interpreting the results. The role of the pharmacist was primarily seen as identifying when pathology for TDM was required and providing advice on the pathology results. These were similar to findings from an earlier study by Norris et al., which reported that, within the MDT, the pharmacist was the most frequently nominated professional accessed to provide dosing and monitoring advice [20], whereas hospital TDM services were most commonly performed by medical practitioners rather than pharmacists [16]. Major issues with current TDM processes in hospital and health services which were highlighted in the survey were inappropriate or no sampling for medication levels and inappropriate or no review and interpretation of the results and future medication doses. This would present an opportunity for greater pharmacist involvement [5,28,29]. Pharmacists are recognized as the medication experts and are ideally placed to perform TDM in specific circumstances, given their knowledge of pharmacology, pharmacokinetics, and concentration-effect relationship [1,8,21]. The survey showed a trend of Australian hospitals moving towards digital systems (or integrated electronic medical records), with most hospitals reporting a complete digital system or a hybrid of a paper-based and a digital system. A recent Australian study showed the transition from a paper-based to a digital system (ieMR) had no significant impact on two areas of interest; those being appropriate sample collection for TDM and appropriate dose adjustment. However, the study also showed that, overall, regardless of the type of system, the odds of an appropriate sample being taken for TDM increased with pharmacist involvement [30]. With this impact of pharmacists documented in the literature, there are opportunities in the future for expanding the scope and role of pharmacists in areas including pharmacist-managed TDM programs within the Australian hospital and health service, giving them responsibility for ordering pathology, and prescribing subsequent medication doses based on their clinical review. This is regardless of whether the hospital utilizes an electronic medical record system and whether these systems have streamlined and improved workflows and processes as publicized [16,17].

In this study, approximately 15% of respondents provided a pharmacist-managed TDM service to their health facility. These programs differed in both the medications and role of the pharmacist within the program. The antibiotic medications (aminoglycosides and glycopeptides) were the most common groups monitored by pharmacists, followed by digoxin and the anti-epileptic class of medications. The role of the pharmacist within the TDM service varied across facilities. The most commonly identified roles involved providing advice and the review of pathology, whereas the ordering of medication-related pathology and prescribing subsequent medication doses were rarely mentioned. These roles within the Australian healthcare system have not been fully explored, mainly due to current regulations defining practice [2]. These results contrast with those in the US, where in a 2016 survey, 89.9% of hospitals allowed pharmacists to order serum medication levels and other clinically important laboratory tests, and 86.8% of hospitals allowed pharmacists to write medication orders. In another survey, approximately 18% of community pharmacies reported to be performing clinical laboratory tests, including those for lithium and tricyclic anti-depressants TDM [31,32]. In Saudi Arabia, a recent survey demonstrated that, at 37.84% of hospitals, pharmacists have the ability to request an estimation of patients’ drug levels and, at 30.77% of hospitals, they have the ability to change drug sampling time [18].

Themes of medication safety, resourcing and funding, and governance/socio-political matters were seen in the free-text comments provided by the respondents. More specifically, taking ownership of medication safety, role expansion, and utilizing the skill set of pharmacists were perceived strengths. Increased workload and maintenance (i.e., regular education and credentialling) of the program were identified as weaknesses of such a program and were a significant theme within the free-text section. This issue of workload and resources within the pharmacy workforce is one of significance and should be balanced against the value added by pharmacists. Further work and research could be conducted within this area. Opportunities included the potential for greater engagement with healthcare facilities and the multidisciplinary team. The potential for the deskilling of medical officers was identified as a potential threat, which is pertinent in the current healthcare climate in Australia, with strong discussion around pharmacist prescribing. An overwhelming theme identified by respondents was the need for a policy around TDM and the potential for its scope to include an expanded role for pharmacists.

The themes identified in the survey were similar to those identified in a recent scoping review of pharmacist prescribing in the UK, Canada, New Zealand and Australia (Zhou et al.) [28]. Zhou et al., found there were three major barriers. These were the socio-political context, resourcing issues and prescriber competence. The most common barriers were inadequate training regarding diagnostic knowledge and skills, inadequate support from authorities and stakeholders, and insufficient funding/reimbursement. These barriers to providing TDM services supported a study by Kheir et al. (2015), which raised issues around pharmacists spending most of their time on dispensing medications and inventory issues rather than direct patient-care services, the lack of practical knowledge to implement the basics of pharmacokinetic (PK) principles to provide effective TDM services, and the lack of PK-related continuing education topics and training [16,17].

The study also found TDM services were mostly performed by healthcare practitioners (for example, medical officers), rather than pharmacists, in most hospitals. However, in countries such as Australia, the US and the UK, where pharmacists receive extensive training on pharmacokinetic monitoring and have demonstrable skills, it is the pharmacist who is generally referred to for advice [8]. In an Australian and New Zealand survey conducted in 2010 by Norris et al., the pharmacist was cited as being the most frequently accessed health professional to provide TDM in hospitals [21]. In a survey conducted in Saudi Arabia in 2018, pharmacists were able to request a drug level in 38% of hospitals [18].

The study had both strengths and weaknesses. A strength of the study was the utilization of an online platform to administer the survey, allowing access to the survey nationally. Limitations included the sample response rate was quite low at 38%. However, this is comparable to other recent pharmacist surveys looking at service provision both within Australia and internationally. A survey investigating pharmacists’ perceptions of vancomycin area under the curve (AUC) monitoring in Australian hospitals reported a response rate of 43%; a pre- and post-pharmacist survey on tobramycin dose recommendations in an Australian hospital reported a response rate of 30% and 23.9%, respectively; while a Polish survey of pharmacists’ knowledge, attitudes and barriers in pharmaceutical care reported a response rate of 44% [14,33,34,35,36].

Although the response rate was small, a strength of the study was that the survey captured a broad range of pharmacy departments. The respondents were from all states and territories, and from metropolitan and rural areas, and included private and public hospitals.

Another limitation of this study was that there were some missing data due to respondents not answering all the questions. Although the missing data were small in scale and are a common problem, especially in questionnaire-based population surveys, it may have led to bias. 

## 5. Conclusions

Based on the findings of this survey, there is some evidence of pharmacist-managed TDM models operating within Australian hospitals. These models differed between hospital sites. Effective medication and the expanding roles of pharmacists must be supported by professional training, and legal and regulatory frameworks. In an evidence-based health system, it is important to show evidence of the benefits and impacts to support these changing roles and perceptions of the pharmacist [36]. If the implementation of pharmacist-managed TDM programs is to occur, including roles such as medication-related pathology ordering and medication dose prescribing, then identified barriers, including stakeholder engagement and pharmacist competence, must be addressed. As such, a concentrated effort is required to develop clear policy pathways, including targeted training courses, raising stakeholder recognition, and identifying specific funding, infrastructure and resourcing needs to ensure the smooth integration of these programs and these additional roles for pharmacists into hospital and healthcare systems.

## Figures and Tables

**Table 1 pharmacy-10-00135-t001:** Demographic Characteristics.

**Respondents per State or Territory *n* = 92**	
Victoria	37% (34)
New South Wales	28% (26)
Queensland	21% (19)
Western Australia	6% (5)
South Australia	3% (3)
Australian Capital Territory	2% (2)
Tasmania	2% (2)
Northern Territory	1% (1)
**Hospital Characteristics *n* = 92**	
Hospital Characteristics	
*Classification*	
Public Hospital (Metropolitan)	42% (39)
Public Hospital (Regional/Rural/Remote)	42% (39)
Private Hospital Metropolitan	16% (14)
*Number of Acute Inpatient Beds*	
<100	16% (15)
101–500	51% (47)
501–1000	33% (30)
*Integrated Electronic Medical Record*	
Yes	39% (36)
No	37% (34)
Combination of Both	24% (22)

**Table 2 pharmacy-10-00135-t002:** Main Roles within the TDM Process (*n* = 92).

	Pathology Ordering (*n*)	Review and Interpretation of TDM Results (*n*)
Medical Officers	80% (74)	50% (46)
Nurses	4% (4)	12% (11)
Pharmacists	13% (12)	32% (29)
Other	2% (2)	7% (6)

**Table 3 pharmacy-10-00135-t003:** Pharmacist Perceptions of Current Issues within the TDM process ^¥^.

	*n* = 92	Average Response	Standard Deviation
Inappropriately timed sample collection	39 (42.4%)	3.2	0.8
No sample being collected	36 (39.1%)	3.0	0.8
No actioning of results from therapeutic drug monitoring assays	36 (39.1%)	2.9	0.8
Inappropriate actioning of assay result (i.e., incorrect dose calculation	36 (39.1%)	2.7	0.7
Not Completed ^≠^	56 (60.8%)		

^≠^ Indicates respondents did not answer these questions. ^¥^ Able to select more than one response.

**Table 4 pharmacy-10-00135-t004:** Medications within current pharmacist-managed TDM programs within Australia (*n* = 13).

Medication	Monitored in Pharmacist TDM Program
Aminoglycosides	76% (10/13)
Glycopeptide Antibiotics	76% (10/13)
Digoxin	23% (3/13)
Anti-epileptics	15% (2/13)
Clozapine	8% (1/13)
Warfarin	8% (1/13)

**Table 5 pharmacy-10-00135-t005:** Pharmacist Responsibilities within a Pharmacist-managed TDM Program.

	*n* = 13	Average Response	Standard Deviation
Recommendations when TDM is required (to medical staff)	10 (76%)	4.0	0.8
Ordering relevant pathology for TDM	10 (76%)	1.9	1.4
Alerting medical staff when TDM results are available	10 (76%)	3.3	1.1
Recommendation’s post TDM results including any changes in dose, frequency for subsequent doses	10 (76%)	4.2	0.8
Prescribing subsequent doses	10 (76%)	1.9	1.5
Not Completed ^≠^	3		

^≠^ Indicates respondents did not answer these questions.

**Table 6 pharmacy-10-00135-t006:** Education or Credentialling Package delivered by pharmacist-managed TDM program.

	*n* = 13	Average Response	Standard Deviation (SD)
Self-Directed Learning	10 (76%)	3.4	1.9
Department Clinical Educator	10 (76%)	2.0	1.6
Senior/Specialist Pharmacist	10 (76%)	2.9	1.7
External Person	10 (76%)	1.0	0.0
Not Completed ^≠^	3		

^≠^ Indicates respondents did not answer these questions.

## Data Availability

The data presented in this study are available on request from the corresponding author. The data are not publicly available due to privacy.

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
