# Peer review of "Pharmacist-Managed Therapeutic Drug Monitoring Programs within Australian Hospital and Health Services—A National Survey of Current Practice"

_pharmacy, 2022, doi:10.3390/pharmacy10050135_

Round 1
Reviewer 1 Report
Thank you for the opportunity to review the manuscript for ‘Pharmacist managed therapeutic drug monitoring programs within Australian hospital and health services - A national survey of current practice’.
The manuscript reads well, and only minor changes are suggested. My comments to improve the manuscript:
Abstract:
Should TDM be defined at first use?
I’m not sure the background/method lines of abstract set out the research aim. The last line of the background seems to pre-emptively address the research aim. Perhaps, rephrase setting out the unknown.
Methods- include the setting
Results- I find the second and third lines of results contradictory, are pharmacists highly involved, if only applicable to 15% of respondents? Perhaps, it is in the minority of hospitals, where there is a pharmacist managed service, areas they are involved in are….
Intro L42- the information is for primary care in US, any info on secondary care?
Methods: clear
Results;
Did type of hospital public/private or digital maturity have an effect on TDM roles for pharmacists?
Themes section is a little underdeveloped, if word count permits, suggest fleshing this out, and also inclusion of illustrative quotations.
Discussion: well thought out and considered. One suggestion- perhaps call out the relative advantage of pharmacist managed services, if the baseline is mostly medical managed. Barriers are described e.g. sufficient PK knowledge, so why should we allow pharmacists manage them?
Reviewer 2 Report
The authors describe the results of a survey of TDM services within Australian hospitals. The manuscript is well detailed and with the supplement, provides evidence to an area of opportunity for pharmacist to optimize patient care through enhancing and starting TDM services. A few comments:
To this reviewer, the word "pathology" in line 34 is unclear. Does this mean drug levels? Perhaps, this is a common term used in Australia, but this reviewer from the US needs a little help.
The TDM of specific drug classes asked within the survey should be within the manuscript methods. While this was in the supplement, it is also detailed in the results and should be listed within the methods section.
I may have missed this but were any other drugs listed besides those that were selected in the survey? For example, did any respondents suggest the have beta-lactam TDM monitoring?
The 3. from results on line 137 needs a return (editing more likely than reviewer action).
MDT is not defined on line 190, but is on line 210. Need to switch.
Reviewer 3 Report
The aim of the manuscript is to identify the extent to which pharmacist managed therapeutic drug monitoring (TDM) services exist within Australia and the roles and responsibilities of the pharmacists involved. A cross-sectional online survey was conducted, which explored the presence of such a program, the scope of practice, operational requirements, attitudes and practices of pharmacists toward the service. The survey also explored any perceived barriers and enablers associated with the establishment and delivery of these services. Analysis was performed using the Stata statistical software package.
In general, the structure and concise working of the manuscript are adequate, which greatly facilitates its understanding by potential readers. On the other hand, tables and figures in the manuscript are adequate.
Comments: The Authors are kindly invited to include a definition of TDM in the Introduction Section. How do hospitals evaluate drug concentrations? Do they use a PK software? What criteria do they use to change the dose?
